# Development of Embroidery-Type Pressure Sensor Dependent on Interdigitated Capacitive Method

**DOI:** 10.3390/polym14173446

**Published:** 2022-08-23

**Authors:** TranThuyNga Truong, Ji-Seon Kim, Jooyong Kim

**Affiliations:** Department of Smart Wearables Engineering, Soongsil University, Seoul 156-743, Korea

**Keywords:** capacitive pressure sensor, electro-textile, wearable sensor, dielectric, embroidery-type sensor

## Abstract

Many studies have been conducted to develop electronic skin (e-skin) and flexible wearable textiles which transform into actual “skin”, using different approaches. Moreover, many reports have investigated self-healing materials, multifunctional sensors, etc. This study presents a systematic approach to embroidery pressure sensors dependent on interdigitated capacitors (IDCs), for applications surrounding intelligent wearable devices, robots, and e-skins. The method proposed a broad range of highly sensitive pressure sensors based on porous Ecoflex, carbon nanotubes (CNTs), and interdigitated electrodes. Firstly, characterizations of ICDs embroidering on a cotton fabric using silver conductive thread are evaluated by a precision LCR meter throughout the frequency range from 1 kHz to 300 kHz. The effect of thread density on the performance of embroidered sensors is included. Secondly, the 16451B dielectric test fixture from Keysight is utilized to evaluate the composite samples’ dielectric constant accurately. The effect of frequency on sensor performance was evaluated to consider the influence of the dielectric constant as a function of the capacitance change. This study shows that the lower the frequency, the higher the sensitivity, but at the same time, it also leads to instability in the sensor’s operation. Thirdly, assessing the volume fraction of CNTs on composites’ properties is enclosed. The presence of volume portion CNTs upgrades the bond strength of composites and further develops sensor deformability. Finally, the presented sensor can accomplish excellent performance with an ultra-high sensitivity of 0.24 ${\mathrm{kPa}}^{-1}$ in low pressure (<25 kPa) as well as a wide detection range from 1 to 1000 kPa, which is appropriate for general tactile pressure rages. In order to achieve high sensor performance, factors such as density, frequency, fabric substrate, and the structure of the dielectric layer need to be carefully evaluated.

## 1. Introduction

In recent years, the demand to monitor physiological data for wearable health sensors, particularly textile sensors, has become an exciting issue and attracted significant research interest. Among these sensors, the exceptional properties of pressure sensors make them a promising component in the next generation of flexible electronics. Electronic skin and wearable gadgets are two significant parts of adaptable electronic innovation [1,2,3]. The advancement of electronic skin is persuaded by developing interest in computerized reasoning, human-machine interfaces, and artificial skin. Additionally, flexible wearable gadgets have extraordinary potential for the user in fundamental features of health and nursing applications. Wearable sensors, particularly smart textile sensors using electro-textile, have become an intriguing issue and have drawn critical interest from specialists. The combination of the Internet of Things (IoT) technology and intelligent textiles has led to the growth of the smart textile market. Many efforts have been studied to develop flexible pressure sensors. Many approaches for measuring pressure depend on piezocapacitive, piezoelectric, triboelectric, and piezoresistive effects. Among them, capacitive pressure sensors depending on a parallel plate capacitor are widely used due to their lower power consumption, faster response times, and simple structure. Capacitive sensors are defined by the fact that the capacitance changes under external stimulation. A typical capacitive sensor is a parallel plate capacitor, which is usually separated by an insulating dielectric layer sandwiched between upper and lower conductive electrodes. In theory, the capacitance of a parallel plate capacitor is given by the Formula (1):(1)$$C=\frac{\varepsilon_{r}\varepsilon_{0}A}{d}$$ where $\varepsilon_{r}$ represents the dielectric constant of the material, $\varepsilon_{0}$ vacuum permittivity is, *A* is the effective area of upper and lower plates, and *d* is the thickness or spacing between two electrodes. It can be seen that the capacitance is affected by three factors: the surface area of the two conductive plates (*A*), the distance or gap d between the two electrodes (*d*), and the relative permittivity of the middle dielectric layer (*ε*). By changing $\varepsilon_{r}$, *A*, and *d*, the capacitive sensors can be divided into three types: variable dielectric, variable area, and variable spacing distance [4,5,6,7,8,9]. In these approaches, under an external mechanical force, the geometric shape between the two conductive electrodes varies, leading to variance in the surface area and the distance. As a result, simultaneously, the capacitance also changes. This is the variable area and distance type. Due to the change in space (*d*), the network between the dielectrics also changes, resulting in a difference in the dielectric constant, called the variable dielectric type. For capacitive pressure sensors using the parallel method, a higher thickness, more extensive surface, and dielectric layer achieve more variety in capacitance. This means this type of pressure sensor depends on the size to have high performance. Furthermore, due to the structure of the parallel type, such a sensor’s design consists of two electrodes placed facing each other, making it difficult to attach to clothing or textiles. In this work, we propose the designing and implementing of a textile pressure sensor based on calculating interdigital capacitance. We focus only on the modification of the dielectric layer to improve sensitivity. This proposed method uses only one electrode on one side of the fabric, so the sensor is not affected by the distance of the dielectric layer but instead detects a variable capacitance from changing frequency and dielectric of the porous polymer layer under compression. The electrodes are fabricated using conductive silver thread embroidered on cotton fabric, which resembles the comb with multiple interdigitated fingers. They were evaluated by adjusting embroidery thread type, angle, and density factors. Moreover, to enhance the sensitivity of flexible capacitive pressure sensors, graphene nanoplatelets (GNP) and carbon nanotubes (CNTs) mixed with different proportions inside elastomers such as polyurethane (TPU) are utilized as dielectric materials [10,11]. Other types of composites, for example PDMS and ecoflex, also play essential roles in the design of creating flexible pressure sensors to make porous, sponge-like, and foam-like composites that are easy to cure at average room temperature [12,13]. In addition, a low-cost flexible capacitive pressure sensor has been proposed, which has highly flexible polymeric foam as the dielectric [12]. Another method used extensively in recent studies to enhance the sensitivity of flexible pressure sensors is the wrinkled structure [14,15,16,17]. This technique is usually utilized in both capacitive and resistive sensors. After fabrication, wrinkled structures have been created that work as sensing dielectric material.

The efforts to increase the sensitivity in this paper are grouped into two major studies: the dielectric change in the composite layer and the effect of frequency on the sensor’s performance. Finally, the experiment results show that the proposed sensors could be advantageous in size, sensitivity, cost, durability, and power consumption, which will have many potential applications in the next generation of wearable electronics.

The rest of the paper is separated into the following sections. Firstly, the experiment introduces novel pressure techniques based on changing the relative permittivity. Secondly, “fabrication” illustrates the fabrication process of the proposed capacitive pressure sensor. Next, “measurement results and discussion,” where the comparison of characteristics under effecting by pressure, including effects of frequency, sensitivity, cost, and durability. Finally, conclusions are drawn in the last section.

## 2. Materials and Methods

### 2.1. Conductive Tracks and Principle of the Transducer

The interdigitated capacitor utilizes lumped circuit components known as a multi-finger periodic construction. The sensor is fabricated based only on one port coplanar interdigital capacitor (IDC), so it requires only one size connecting from sensor to controller directly. In contrast to the parallel plate capacitor, the interdigitated capacitor requires only unevenly identifying varieties of material under test (MUT). Moreover, the principal benefits of selecting one port coplanar allow for the achievement of miniaturization and experimentation simpler and faster. The design has a better quality factor than a parallel capacitor [18]. In this design, the capacitance sensing area is loaded between slender fingers holes. The material under test (MUT) is applied on integrated fingers. Here, this MUT is known as a composite dielectric layer with different relative permittivity values when pressure is applied. When the gaps between fingers move closer, the capacitance increments likewise. The state of the sensor is portrayed by the boundaries displayed in Figure 1. Typically, we selected the element of the holes among fingers (G), and spaces toward the finish of fingers are equal. The capacitor consists of ten fingers displayed in Figure 1 and has the accompanying boundaries shown in Table 1. Due to the potential for high conductive, low resistance, and minimal expense, the design of the introduced sensor used silver conductive thread as a conductive material, which means the fabric substrate is directly attached by embroidery.

When the MUT (material under test) is applied to the interdigital capacitor electrodes, the capacitor across the finger electrodes will change due to the frequency and dielectric variations. Finally, the presented sensor transforms dielectric changes of MUT into pressure. The capacitance change for the interdigital microstrip structure is determined by summing up the unit cell capacitance (Figure 1b). Each unit cell is calculated as Formula (2) [19]:(2)$$C_{Cell}=C_{MUT}+C_{Sub}+C_{G}$$
(3)$$C_{MUT}+C_{Sub}=\varepsilon_{0}\frac{\left(\varepsilon_{\mathrm{MUT}}+\varepsilon_{\mathrm{Sub}})K(\sqrt{1-\delta^{2}}\right)}{2K\left(\delta \right)}$$
(4)$$C_{G}=\varepsilon_{0}\varepsilon_{\mathrm{MUT}}\frac{h}{a}$$
(5)$$\delta =\frac{h}{a}$$
where $\varepsilon_{0}=8.85\times 10^{-12\;}F/m$ is the relative permittivity of free space, $C_{Sub}$ is the capacitance of substrate, and $\varepsilon_{MUT}$ is the relative permittivity of the MUT. $C_{MUT}$ is the capacitance of material under test (MUT). $C_{G}$ is the capacitance between electrodes. *K(x)* is the elliptic integrals of the first kind, h is the thickness of the metal layer, and a is the dimension of one unit cell.

### 2.2. Methods

Dielectric measurements were calculated with the 16451B dielectric test fixture from Keysight (Agilent 16451B dielectric fixture) using guarded electrodes following Agilent Technologies Manual and a precision LCR meter (E4980AL). They were utilized to accurately evaluate the dielectric constant of the Ecoflex samples with and without an air gap and CNTs. A non-contact approach was used to perform the measurements. The composites were placed at a lower position among the guard electrodes. In this approach, the upper guard electrode was measured at a particular distance. The thickness of the dielectric to be measured is denoted as $t_{m}$, and the gap formed at this time is denoted as $t_{g}$. In order to obtain the constant dielectric results, two capacitance values are required. First, $C_{s1}$ denotes the capacitance value measured without a dielectric material under test (MUT), and $C_{s2}$ denotes the capacitance after the MUT has been applied to the guard electrodes. The formula for applying this principle can be found in Formula (6), given in the Keysight manual. Figure 2 illustrates the experience measurements.
(6)$$\varepsilon_{r}=\frac{1}{1-(1-\frac{C_{s}1}{C_{s}2})\times \frac{t_{g}}{t_{m}}}$$

In order to study the effect of various amounts of air in the Eco-flex samples, an LCR meter (E4980AL) was set up with 201 points ranging from a frequency of 1 kHz to 300 kHz. The measurements under 5 kHz were unstable; therefore, all experiments were carried out in a frequency range of 5 kHz to 300 kHz. The samples were applied with a pressure tester (Dacell Co., Seoul, Korea) during periods of 5 kHz, 10 kHz, 50 kHz, and 290 kHz.

Three sets of experiments were carried out. In the first one, the samples consisting of three different embroidery densities (Low, medium, and high) are measured to evaluate the baseline capacitance value. This measurement is used to assess the practical capacitive values at different density adjustments without deforming the fabrics. In the second set of tests, the measure of four composites reveals the effect of the presence of CNTs and porous. This ensures whether frequency affects the dielectric constant and, if so, which frequency points should be considered. All samples were measured under this condition. In the third set of experiments, under pressure, we choose the suitable density and proper frequency from the results of the two tests above. In this case, CNTs porous composites were used in sequence, starting from the third test to the end.

### 2.3. Materials

The fabric cotton was used as a substrate for the sample preparation with the conducting electrode to fabricate flexible pressure sensors. Cotton was chosen because of its flexibility and suitability for the human body’s curves. Moreover, by combining the silver thread known as a conductive material with a textile such as cotton, fabricated samples will have a hybrid conductive structure, promoting the capacity to create sensors with larger regions, which is not generally possible with other materials. Each detector electrode consists of two parts: a nonconductive cotton on the top; and a conductive layer embroidered by a machine (brother PR670E, Kaohsiung City, Taiwan) using silver-coated conductive yarn (AMANN silver-tech, Broomfield city, CO, USA) on the bottom (shown in Figure 3).

Silver-coated yarn polyamide cores used in this study are composed of multiple silver-plated nylon fibers (the diameter of monofilament is 20 μm after coating). When a nonconductive core, nylon, is used, the conductive yarn is able to stretch for wearability, which leads to electromagnetic properties of conductive yarns far different from the metallic threads, which have a pure state (99.999%). It is the most suitable choice for embroidery. Silver-coated yarn polyamide cores are all twisted 34 filaments to create a single conductive yarn. We selected this silver-tech conductive yarn due to its high conductivity and low corrosion resistance, which is ideal for conductive surfaces, especially sewing or embroidery [20]. The electrical resistance of each ply of yarn, as determined by an LCR meter, is shown in Figure 4. It is evident that as length increases, resistance increases while decreasing as the number of plies rises. This indicates that the electrode’s conductive wire has a low resistance, making it an appropriate material for use as a capacitance sensor. In addition, since there is no cellular damage in the cytotoxicity test conducted in accordance with DIN EN ISO 10993-5, it can be used in daily life without harming the human body. The embroidery Brother PR670E has a straightforward production process, allowing for achieving high geometrical accuracy on unconstrained placement, and is easily implemented in various shapes. As a result, conductive silver thread and an embroidery machine were used in this study to embroider sensor electrodes directly to make them easy to use and sturdy.

In the weaving process, thread strength reduction is an unwanted effect. The reason is that at high velocity, the surface contact of the silver layer builds the string pressure, which prompts the breaking strength of the conductive string during the interaction. Accordingly, in order to sew conductive layers on the two sides, it needs to reduce the speed of the embroidery process, causing expanding the assembling cost and time utilization. In this paper, the conductive string was utilized exclusively for one side, the base side or bottom side, and the rayon string on the top side, as displayed in Figure 3. Rayon yarn is known as the most common thread utilized in the weaving industry. It has an attractive sheen and is a moderately reasonable price. The average breaking force and elongation graph in [20] indicate that the rayon thread is stronger than the single-ply thread and weaker than the double-ply thread. That is because the elongation of rayon is the shortest, and the single-ply string is the longest, leading to rayon threads stretching more thin, which causes them not to push further, keeping the original capacitor sensor shape. Then the fabricated sensor can deal with the right way with fewer tensions and increasing stabilizing.

Eco-flex (Smooth-on Inc., Macungie, PA, USA) was selected to create the ideal dielectric environment with and without air that was used in the experiment to detect pressure. Eco-flex was chosen as the dielectric because of its low viscosity, making it simple to mix with other materials and causing no surface-level phenomena. It is also safe for the human body and easy to create in complicated shapes. Then, Eco-flex composite is mixed with single-wall carbon nanotubes (CNTs) (TUBALL, diameter smaller than 2 nm, produced by OCSiAl). The reason for choosing CNTs comes from the aspect ratio between the length and diameter of CNTs. The CNTs have a much larger aspect ratio than 1. On the other hand, the spherical particles have an aspect ratio of approximately 1. The aspect ratio of CNTs is a great advantage for lower critical volumes since the percolation threshold reduces with an expanding aspect ratio of the filler [21].

### 2.4. Fabrication

The CNTs microporous dielectric layer or pressure-sensing layer was fabricated using the process shown in Figure 5. The eco-flex solution was obtained by mixing a base (Part A) and a cured agent (Part B) with a volume weight ratio of 1:1. Following this process, granulated brown sugar and CNTs were dispersed in the Ecoflex solution. The different dielectric films were obtained by changing the volume weight ratio of CNTs and Sugar distributed in the Ecoflex, as shown in Table 2. The solution was stirred at 120 rpm for 15 min to help segregate granulated brown sugar and evenly distribute it in the composite. The mixture was cured at room temperature (30 °C) for 3 h. To form the shape of the dielectric films, we used a 3D mold with the length, width, and height of electrodes of the same size (20 mm × 20 mm × 7 mm). Following the curing time, the mixer was removed from the 3D mold, and the sugar was dissolved in boiler water under magnetic stirring at 200 rpm for at least 24 h. Porous Ecoflex Dielectric (ECO-PO) is the sample with air due to the formation of pores. In contrast, non-porous Eco-flex Dielectric (ECO-NPO) was the sample in which no pores were formed because sugar was absent. Similarly, CNT-NPO and CNT-PO are non-porous and porous samples with CNTs. The goal of using brown sugar is to create air holes inside the compound to increase the sensor’s sensitivity, according to Formula (8).

A computerized embroidery machine was used to make the electrodes of the proposed capacitive sensor. The automatic embroidery machine program (PE-design) was used to specify the electrode and connector’s geometry and thread density. The type of fabric used for the black jacket includes 100 percent cotton (woven, 0.22 mm in thickness). Only the upper sheet was embroidered using conductive thread, while the lower sheet was embellished with a regular embroidery thread. Thus, only the shape of the electrodes becomes invisible from the upper side. In other words, the outer surface is the rayon thread side covered with a layer of PU film. PU film was added to create the shield protecting the sensor from environmental disturbances. Figure 2 depicts the manufactured layer structure. Electrodes fabricated by the embroidery machine have different densities of 2, 4, and 6 lines/mm, called low, medium, and high density (shown in Figure 6). In our experiment, when using a high density for fabrication, the distance between the threads on the electrodes’ fingers gets closed, reducing the stability of the capacitive sensor.

## 3. Results

The schematic diagram for evaluating the performance of the presented sensor is seen in Figure 7. The pressure is applied using a Keysight LCR meter (E4980AL) and a force gauge Universal Testing Machine (Dacell Co., Seoul, Korea). They are all connected to a computer to collect the data (Figure 7).

From the first set of experiments mentioned above, we have the results displayed in Figure 8. As can be seen, the higher density, the more capacitance, and the lower the factor loss archived. This is because of the increase in the high layer of electrodes. Increasing silver fiber’s volume (density) also increases conductive layers and expands surface electrode mass density due to the silver thread’s structure [20]. Finally, we tested the relationship between the sensing capacitor and effective density to evaluate the baseline capacitor effect. As results are shown in Figure 8, increasing the initial capacitor leads to a decrease in sensitivity. That is due to variations in capacitance ΔC basing on the initial capacitor $C_{0}$ as shown in Equation (7):(7)$$\Delta C=\Delta \varepsilon C_{0}$$

From (7), a higher baseline capacitance is needed for more changes in capacitance variance. However, the sensitivity of capacitive pressure sensors is calculated by $S=\frac{\frac{\Delta C}{C_{0}}}{P}\;$ where P represents the applied pressure. It is necessary to define that the main challenge to transferring dielectric changes into force comes from Δε. The higher Δε, the more sensitivity can be achieved. As seen in the Figure 9, the low density has the most heightened sensitivity, so from now on, we choose the low-density one for the following experiments.

In this paper, we used two methodologies to develop the deformability of the dielectric layer. The first method depends on the compressive behavior of the porous structure. As explained in some earlier works, the capacitance changes when the dielectric of elastomeric changes [22,23]. In this phenomenon, the non-porous Ecoflex ($\varepsilon_{np}$)’s dielectric constant changed with pressure lower than the porous one, even if its capacitance sensor’s variations are not dependent on the changes in the distance between the two electrodes ($\Delta d_{np})$, as shown in Figure 10. Moreover, the dielectric constant and the distance between the electrodes $(\Delta d_{p})$ vary under pressure more than in the case of a non-porous eco-flex ($\varepsilon_{np}$). Additionally, even under the same pressure, the porous Ecoflex’s capacitance changes significantly, indicating that the shift in capacitance is also substantial [24]. The effective dielectric constant of the porous Eco-flex ($\varepsilon_{r}$), which may be explained by Formula (8) [17,18], increases when the pores are subjected to external pressure. Then the variations in the effective relative permittivity of microporous ($\varepsilon_{r}$) under external pressure can be determined as follow:(8)$$\varepsilon_{r}=\varepsilon_{air}V_{air}+\varepsilon_{Eco-flex}V_{Eco-flex}$$
where $V_{air}$ stands for the volume fraction of air and $V_{Eco-flex}$ for the volume fraction of immaculate Ecoflex, where $\varepsilon_{air}=1$ and $\varepsilon_{Eco-flex}=2.8$ [25]. The compression causes the dielectric layer’s pores to gradually close, which lowers the volume percent of air and raises the volume fraction of Eco-flex. The greater dielectric constant of silicone elastomer ($\varepsilon_{Eco-flex}=2.8$) replaces the lower dielectric constant of pores ($\varepsilon_{air}=1$), increasing the porous Eco-flex composite’s effective dielectric constant [26]. The sensitivity of the capacitance sensor can be increased when air gaps are added to this porous elastomer due to increased deformability because the gaps will make the Eco-flex less stiff and cause more voids to grow inside [24].

The second method is based on percolation theory using CNTs (carbon nanotubes) as a filler load to improve polymer properties. In this method, two main issues should be considered, since they affect the selection of suitable particles. The first is the interfacial interaction between CNTs or reinforcement and polymer, known as the percolation path. The second is the proper distribution and dispersion of CNTs inside the polymer, known as concentrations. The reason for choosing CNTs comes from the aspect ratio between the length and diameter of CNTs. The CNTs have a much larger aspect ratio than 1, leading to smaller critical volumes since the filler’s percolation threshold reduces with an expanding aspect ratio [21]. According to the percolation theory, the relationship between relative permittivity (ε) and the volume fraction of the composite filler can be explained by the following power law [27,28]:(9)$$\varepsilon \;\alpha \;\varepsilon_{p\;}{\left(f_{c\;}-f_{CNTs\;}\right)}^{-t}\;\mathrm{for}\;f_{CNTs\;}<f_{c}$$
where ε is the dielectric permittivity of mixer composites, $\varepsilon_{p}$ is the relative permittivity of polymer, $f_{c}$ is the percolation threshold or critical volume, $f_{CNTs}$ is the volume fraction of filler, and t is the dielectric critical exponent. Increasing the number of CNTs inside the polymer leads to changes in the relative permittivity and elastomeric properties. Therefore, choosing a suitable dielectric elastomer is essential to enhancing the sensitivity, especially under high pressure. Figure 11 shows how CNT-PO can be deformed. Compared to the convenience, the sensor compacted with a porous composite layer is deformed by a much more significant gap ($\Delta d_{p}$ > $\Delta d_{np})$ due to the reducing stiffness. The combination of an increase in the effective permittivity and more extensive deformation increases the sensor’s sensitivity. Therefore, a highly sensitive response can be carried out by applying a porous one.

From the second set of experiments described above, we measured the dielectric of four elastomeric layers. In this experiment, the goal is to demonstrate the effect of presenting non-porous and porous, non-CNT and CNT ones. As shown in Figure 12, when presenting porous, the dielectric of the porous layer is lower than the non-porous one. As shown in Figure 13, the CNT network is immersed inside the ecoflex while CNT plays connections. It is known that due to high aspect ratios and long-range Va der Waals interactions, SWCNTs tend to form ropes or bundles with a highly complicated structure. Therefore, CNTs are spread inside the polymer in the form of bundles, as can be seen in Figure 13, leading to developing bond strength and enhancing the durability of the polymer.

The results also show a relationship between the frequency and the variation in the dielectric constant, especially at the following frequency points: 5 kHz, 10 kHz, 50 kHz, and 290 kHz. They are crucial points used in the next experiments in order to evaluate the effect of frequency on the performance of sensors. The larger gap dielectric between CNT-PO and CNT-NPO occurs in the frequency range below 5 kHz, and the lowest is above 250 kHz.

For an in-depth investigation of frequency, we now move to the last set of experiments. The behavior of the sensing mechanism is shown in Figure 14. As expected, the sensor’s sensitivity when operating below 5 kHz is the largest, while above 290 kHz is the lowest. In addition, in order to investigate the effect of frequency on long-term stability and mechanical durability, we tested the samples during over 5000 compression/release cycles at four frequency points: 5 kHz, 10 kHz, 50 kHz, and 290 kHz. The result is shown in Figure 15. The average values of the samples at four typical selected frequencies 5 kHz, 10 kHz, 50 kHz, and 290 kHz, are 7.66 (standard deviation: 0.473), 7.031 (standard deviation: 0.300), 4.444 (standard deviation: 0.180), and 2.517 (standard deviation: 0.108). As can be seen from the results, the higher the frequency, the more stable and consistent in terms of sensitivity can be. Moreover, the results show that the samples under low pressure were more stable and reversible than those under high pressure. All samples illustrated a rapid recovery time lower than 0.2 s (shown in Figure 16). Two different pressures were applied at low and high compressions, 100 kPa and 400 kPa, respectively, which are higher sensitivities than some previously reported pressure sensors (Table 3). The sensitivity of capacitive pressure sensors is calculated by Formula (10):(10)$$S=\frac{\frac{\Delta C}{C_{0}}}{P}$$
where *P* represents the applied pressure. The higher the density of the conductive fiber, the higher the *C*_0_ value (as shown in Figure 8). Therefore, according to the sensitivity calculation Formula (10), the sensitivity will decrease when the *C*_0_ value increases and vice versa. Table 3 shows that the proposed sensor achieves high sensitivity and wide range compared to the two main types of sensors, capacitive and resistive.

## 4. Conclusions

As a result, in this research, we proposed an embroidery capacitance sensor which has a high sensitivity and stability over a broad range of up to 1000 kPa depending on interdigitated structure and a porous conductive dielectric composite. Accordingly, the presented sensor can accomplish excellent performance with an ultra-high sensitivity of 0.24 ${\mathrm{kPa}}^{-1}$ in low pressure (<25 kPa) as well as a wide detection range from 1 to 1000 kPa, which is appropriate for general tactile pressure ranges. The dielectric layer can achieve outstanding durability due to the highly resilient characteristics of Ecoflex and CNTs. Moreover, the effect of density on the performance of sensor sensitivity plays a key role in insensitivity and detection of signal output. Therefore, selecting a suitable density silver thread for embroidering should be considered. Increasing the sensor’s thread density also affects the embroidery machine’s accuracy. The samples working under low pressure are more stable and reversible than those under high pressure. In addition, frequency also has a significant influence on the performance of the sensor. This study shows that the lower the frequency, the higher the sensitivity, but at the same time, it also leads to instability in the sensor’s operation. This is even more evident when the sensor operates continuously under high pressure. Therefore, in order to achieve high sensor performance, factors such as density, frequency, fabric substrate, and structure of the dielectric layer need to be carefully evaluated. Remember that these factors are all interrelated. The density of the conductive fiber and the sensor’s operating frequency affects the stability, while the fabric substrate and the dielectric layer play a decisive role in the sensitivity.

## Figures and Tables

**Figure 1 polymers-14-03446-f001:**
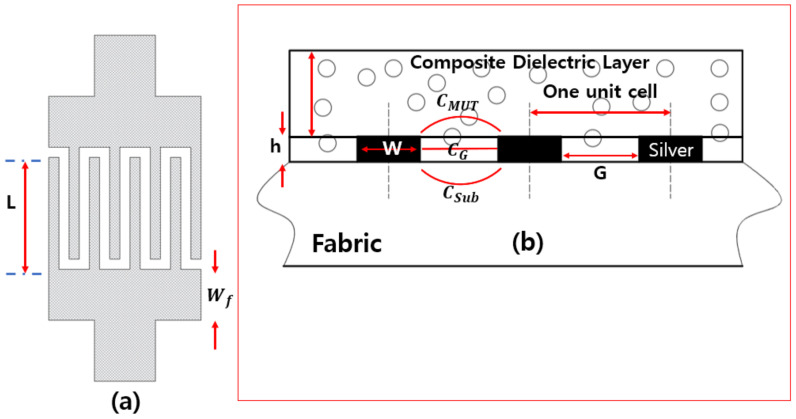
(**a**) Top view and (**b**) cross-section view of the integrated capacitor.

**Figure 2 polymers-14-03446-f002:**
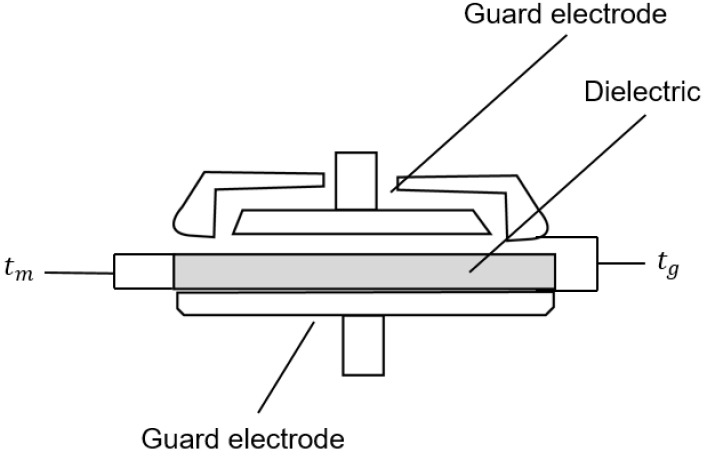
Non-contacting electrode method.

**Figure 3 polymers-14-03446-f003:**
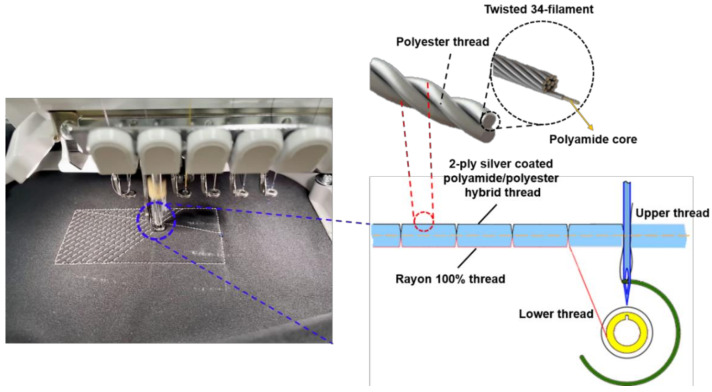
Embroidery process.

**Figure 4 polymers-14-03446-f004:**
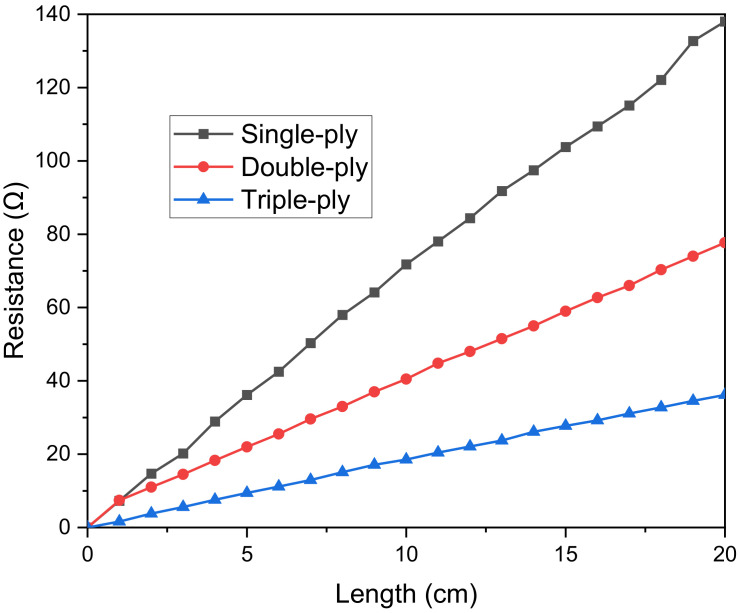
Effect of single-ply, double-ply and triple-ply thread length on the resistance value.

**Figure 5 polymers-14-03446-f005:**
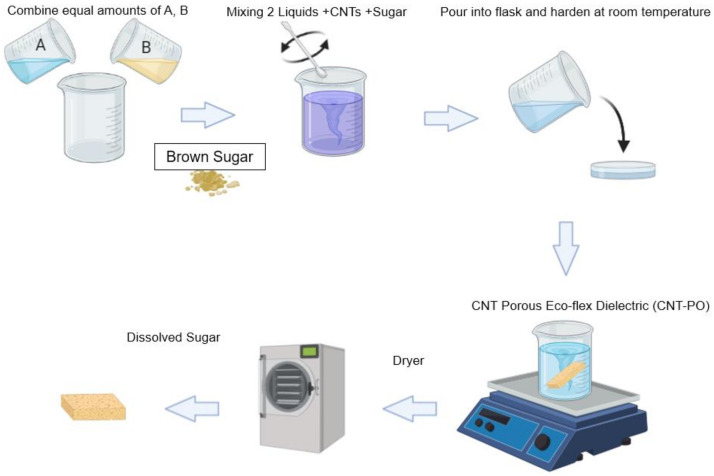
A fabrication process of the Porous Composite Dielectric Layer.

**Figure 6 polymers-14-03446-f006:**
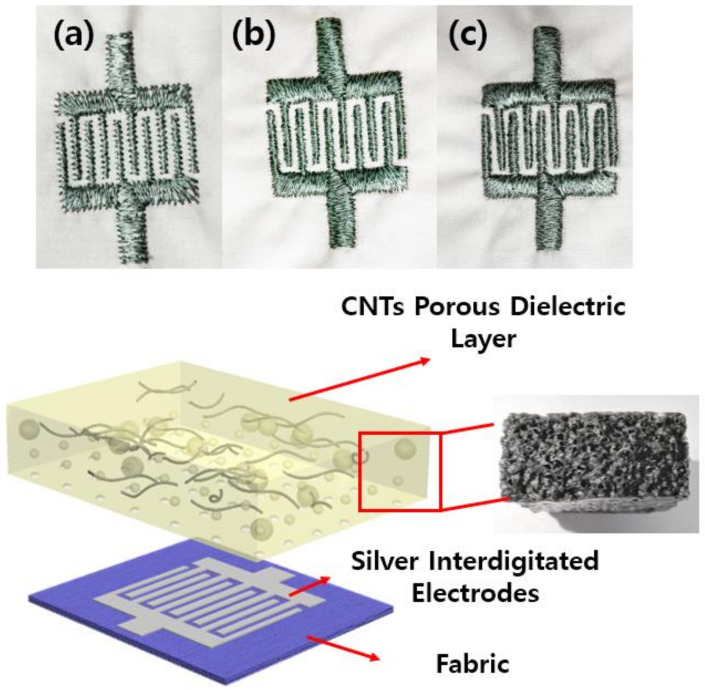
Schematic diagram of the fabricated pressure sensor (**a**) Low density, (**b**) medium density, (**c**) high density.

**Figure 7 polymers-14-03446-f007:**
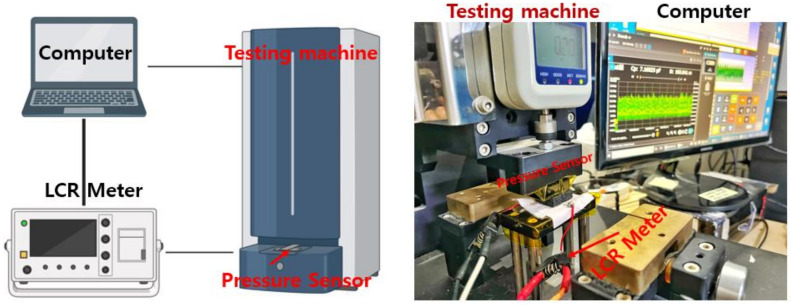
Schematic of the universal testing machine and measurement setup for presented pressure sensor.

**Figure 8 polymers-14-03446-f008:**
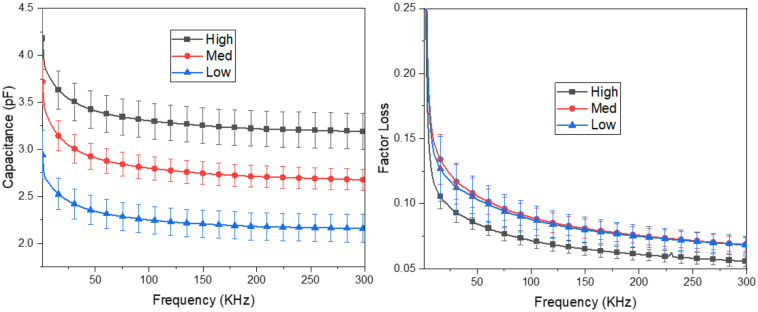
Frequency dispersion of capacitance and loss tangent of initial sensors (low, medium, high density).

**Figure 9 polymers-14-03446-f009:**
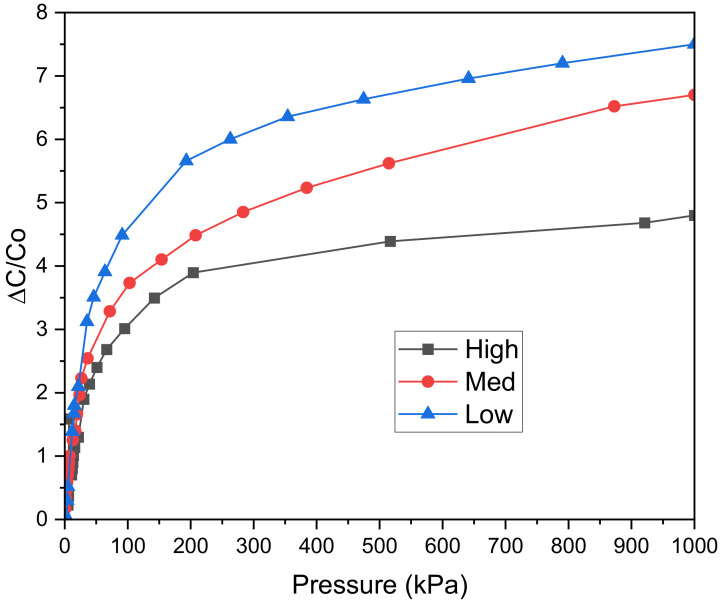
Pressure-response curves of capacitive pressure sensors with different densities.

**Figure 10 polymers-14-03446-f010:**
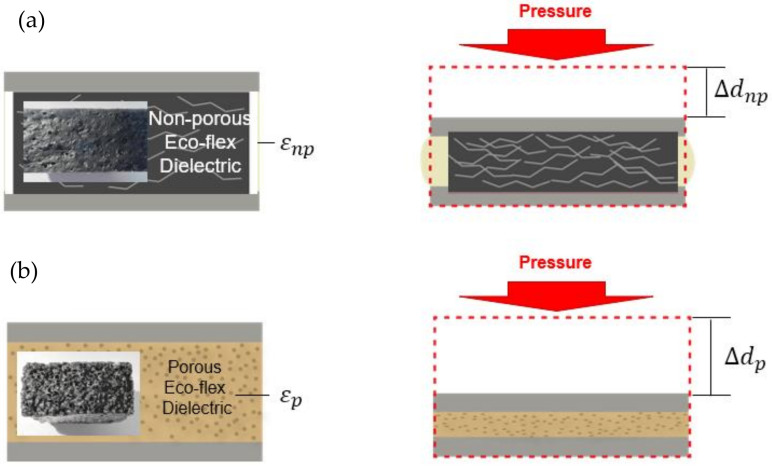
Schematic illustrations of the proposed pressure sensor (**a**) CNTs + Ecoflex; (**b**) CNTs + Sugar + Ecoflex.

**Figure 11 polymers-14-03446-f011:**
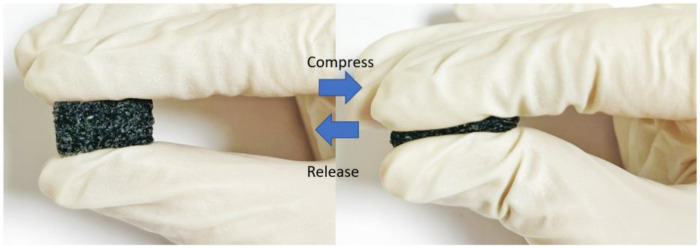
High flexibility of the dielectric layer.

**Figure 12 polymers-14-03446-f012:**
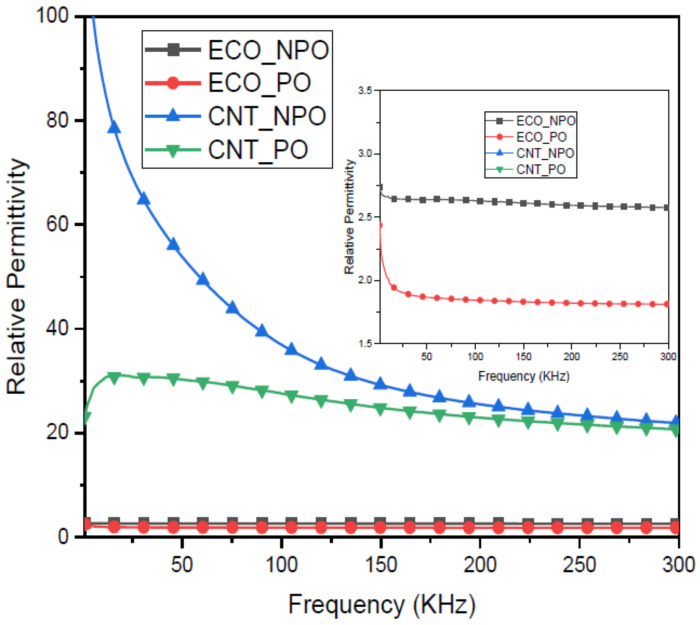
Frequency dispersion of relative permittivity of dielectric layers.

**Figure 13 polymers-14-03446-f013:**
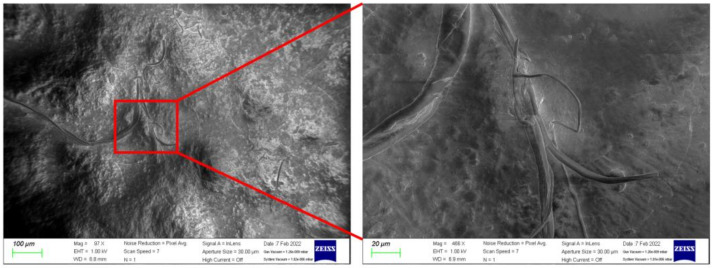
SEM images of the CNTs Composite Dielectric Layer (CNTs + Ecoflex).

**Figure 14 polymers-14-03446-f014:**
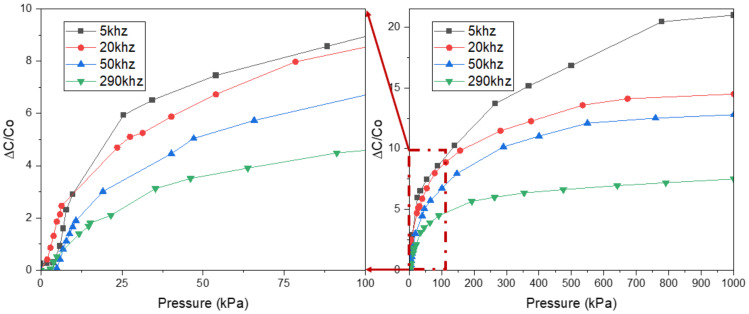
Pressure-response curves of capacitive pressure sensors (low density) at four frequencies.

**Figure 15 polymers-14-03446-f015:**
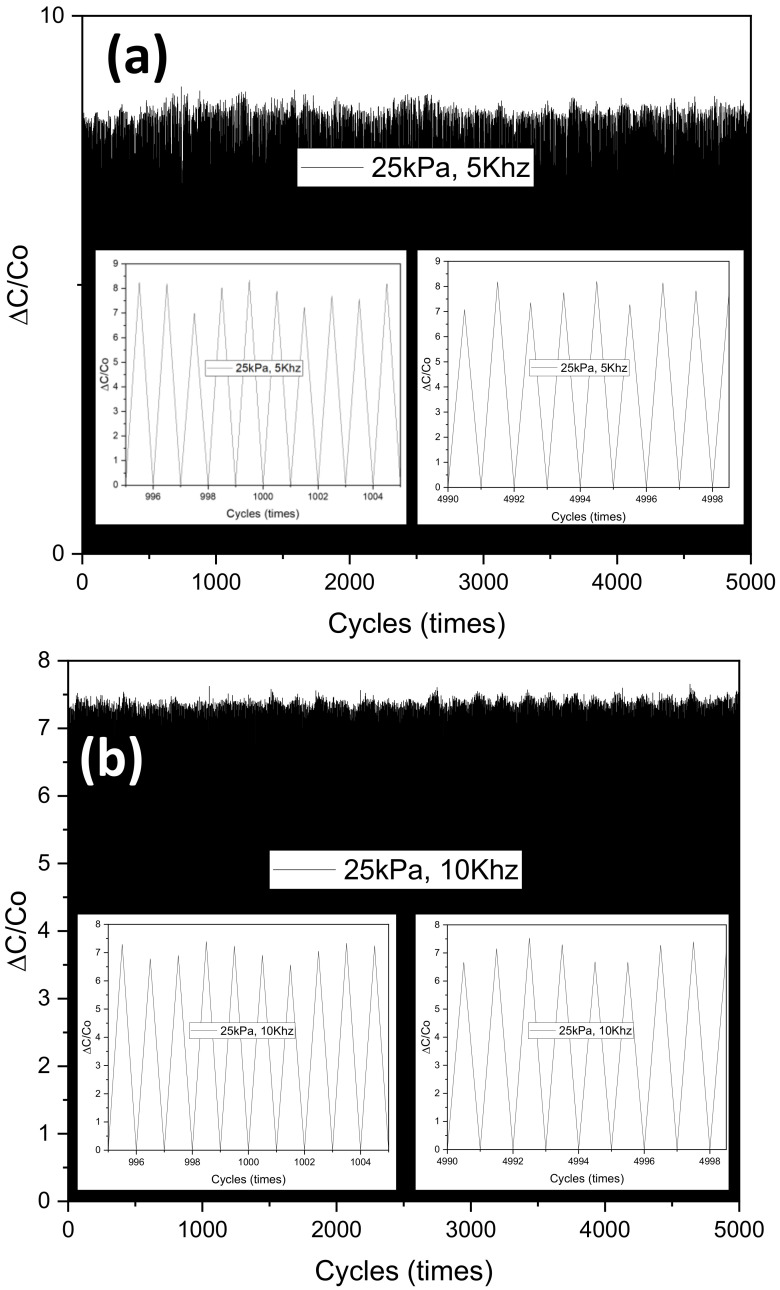
Capacitance response of pressure sensor during 5000 loading and unloading cycles at an applied pressure of 25 kPa (**a**) 5 Khz, (**b**) 10 Khz, (**c**) 50 Khz, and (**d**) 290 Khz.

**Figure 16 polymers-14-03446-f016:**
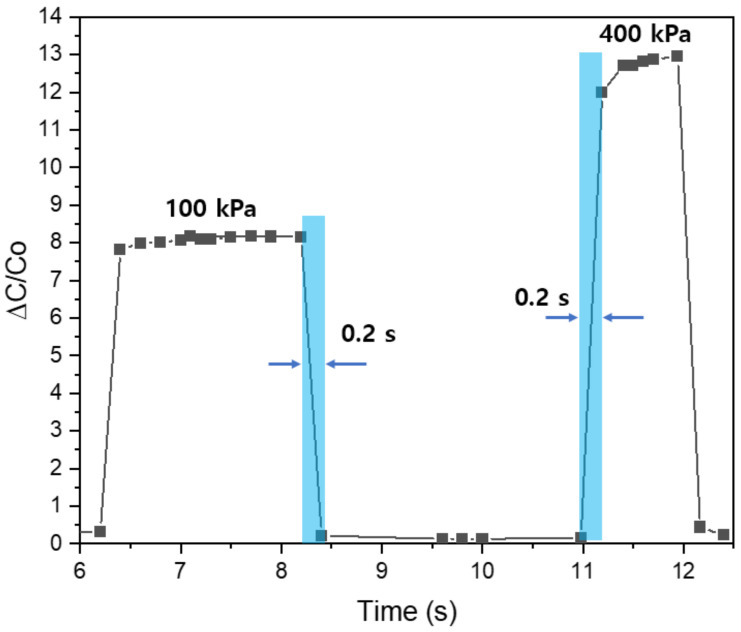
Pressure-response time of capacitive pressure sensors at 100 kPa and 400 kPa.

**Table 1 polymers-14-03446-t001:** Final structural dimensions of the proposed sensor.

Parameter	Dimension
Finger width (W)	1 (mm)
Gap between fingers (G)	1 (mm)
Length of the overlapped region (L)	1.2 (mm)
Width of feedline $\left(W_{f}\right)$	3.5 (mm)

**Table 2 polymers-14-03446-t002:** Dielectric films with different volume ratios of CNTs inside the Ecoflex polymer.

Samples	Volume of Ecoflex (g)	Volume of CNTs (g)	Volume of sugar in the mixture (g)	Percentage of sugar in the Ecoflex (%)	Percentage of CNTs in the Ecoflex (%)
CNTs + Ecoflex	1	$2.5x10^{-3}$	0	0	0.25
CNTs + Ecoflex + Sugar	1	$2.5x10^{-3}$	1	100	0.25
Ecoflex + Sugar	1	0	1	100	0

**Table 3 polymers-14-03446-t003:** Comparison of the proposed sensor with reported pressure.

Reference	Principle	Sensitivity $kPa^{-1}$	Pressure Point (kPa)
This work	Capacitive	0.24 0.022	25 1000
[29]	Capacitive	0.17	1
[30]	Capacitive	0.0121	100
[31]	Capacitive	0.0078–0.24	1–100
[26]	Capacitive	0.18	5
[32]	Resistive	7.62 0.14	50 200
[33]	Resistive	5.67	0.42

## Data Availability

The data presented in this study are available on request from the corresponding author.

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
