# Peer review of "Development of Embroidery-Type Pressure Sensor Dependent on Interdigitated Capacitive Method"

_polymers, 2022, doi:10.3390/polym14173446_

Round 1
Reviewer 1 Report
The paper discusses the fabrication and characterization of a flexible pressure sensor based on capacitive method. They presented the analysis of using silver wire and ecoflex as the flexible dielectric to achieve sensing. Overall, the paper was well organized and clear. A few comments and suggestions for the authors.
In Section 2.4:
-How can you ensure even distribution between the particles? Why is brown sugar added and how is that structure different in shape and size from the CNT particles.
-Wouldn’t most of the solution particles settle at the bottom during the curing phase of the ecoflex material?
-Figure 5 suggests “black sugar” how is that different from brown sugar
Section 3
-Silver may have a high conductivity if its in its pure state (99.999%) but would be very weak to high forces and is prone to breaking. It is also expensive. Did the authors consider different type of material?
Figure 15:
-It is unclear where the snaps are taken from. Very blurry.
Overall comments
-Figures can be better improved. Figure 11 and figure 12 are very large. The authors should be more consistent by using the same font size and type for all.
Author Response
We are thankful to the reviewer for all the valuable feedback.

Reviewer 2 Report
This paper describes a topic on the approach to embroidery pressure sensors dependent on interdigitated capacitors (IDCs) for applications surrounding intelligent wearable devices, robots, and e-skins. The authors proposed a pressure sensors based on porous Ecoflex, carbon nanotubes (CNTs), and an interdigitated electrode. This study shows that the lower the frequency, the higher the sensitivity, but at the same time, it also leads to instability in the sensor's operation. The presence of volume portion CNTs upgrades the bond strength of composites and further develops sensor deformability. Finally, the presented sensor can accomplish excellent performance with ultra-high sensitivity of 0.24 KPa−1 in low pressure (< 25 kPa) as well as a wide detection range from 1 to 1000 kPa, which is appropriate for general tactile pressure rages.
1. It is strongly recommended that the authors should mention clearly the newly developed and /or found point of in section introduction, compared with published papers already reported about pressure sensors combined with CNTs.
2. The authors should compare clearly what the difference for the pressure sensors combined with CNTs in Introduction part, how to effect the performance or reliability mechanisms?
3. The authors described that to achieve high sensor performance, factors such as density, frequency, fabric substrate, and structure of the dielectric layer need to be carefully evaluated. Which one is major factor?
4. The authors should tell readers how to define the quantitative data of sensitivity.
5. The authors described in Table 3 to compare with the proposed sensor containing sensitivity and pressure point. What the major advantage in your work?
6. The authors described the effect of density on the performance of sensor sensitivity plays a key role in insensitivity and detection of signal output. The density of sensing materials always play a key role in sensing performance, right?
Author Response

(The authors gave the same response as above.)
